Effect of harvest on the agronomic, mineral and antioxidant profile of three oregano species (Origanum onites L., Origanum vulgare L. ssp. hirtum, and Origanum acutidens (Hand.-Mazz.) Ietswaart)

Uskutoğlu Tansu tansuuskutoglu@gmail.com
Agriculture Faculty, Department of Field Crops, Pamukkale University , Denizli , Civril , Türkiye
Dąbrowski Piotr
Electronic publication date: 2025 Oct 21
Publication date: 2025
Volume: 13
Electronic Location ID: e20223
Received 2025 Mar 12; Accepted 2025 Sep 22
Copyright: ©2025 Uskutoğlu
Copyright year: 2025
Copyright holder: Uskutoğlu
License: This is an open access article distributed under the terms of the Creative Commons Attribution License, which permits unrestricted use, distribution, reproduction and adaptation in any medium and for any purpose provided that it is properly attributed. For attribution, the original author(s), title, publication source (PeerJ) and either DOI or URL of the article must be cited.
License URL: https://creativecommons.org/licenses/by/4.0/

Keywords: Heavy metals, Micro nutrition, Macro nutrition, Antioxidants, Harvest time, Oregano

Funding: The author received no funding for this work.

==============================
This study primarily aims to examine the variations in plant characteristics, micro- and macronutrient levels, heavy metal content, and antioxidant properties (2,2-diphenyl-1-picrylhydrazyl (DPPH), ferric reducing antioxidant power (FRAP), and 2,2’-azino-bis(3-ethylbenzothiazoline-6-sulfonic acid) (ABTS)), along with antioxidant compounds such as phenolics and flavonoids, across different oregano species cultivated under identical ecological conditions. Origanum vulgare demonstrated the highest fresh (292.25 g/plant) and dry herbage yield (153.34 g/plant). The antioxidant activity varied significantly depending on the oregano species and harvest time. Origanum acutidens showed the highest DPPH radical scavenging activity during the early flowering stage. Conversely, Origanum onites L. displayed the lowest antioxidant activities, as measured by DPPH assay, during the same developmental phase. O. vulgare exhibited the strongest ABTS activity (0.997 mg TE/g dry weight (DW)) in the early flowering stage. Antioxidant activity in O. onites tended to decrease from pre-flowering to full flowering across all assays. Total phenolic content followed a similar trend, with O. vulgare reaching a maximum value at early flowering. Total flavonoid content was highest in O. acutidens during later development stages. The most abundant macronutrients, based on their average values of different harvest times, were calcium (Ca), potassium (K), magnesium (Mg), and phosphorus (P), respectively. The micro, macronutrient, and heavy metal content varied significantly between oregano species and across harvest periods within the same species. Heavy metal levels in oregano samples generally fell below WHO safety limits, except for cobalt (Co) and Chromium (Cr). Multivariate statistical techniques, including principal component analysis (PCA) and correlation analysis, were utilized to elucidate the interrelationships among oregano species, plant characteristics, antioxidant capacity, and mineral composition, thereby providing a comprehensive assessment of the factors governing oregano’s qualitative and nutritional characteristics. These findings emphasize the critical role of oregano species and harvest timing in determining plant growth, antioxidant characteristics, and nutritional properties.

INTRODUCTION

Türkiye has a very diverse range of plants in terms of Thymus (42 species and 47 taxa) (Ağalar et al., 2021), Origanum (21 species, 24 taxa) (Arabaci et al., 2021), Satureja (16 species) (Carikci et al., 2020), and Thymbra (two species, four taxa) species (Kerem et al., 2023). In the Turkish flora, a diverse array of these species holds significant importance. The use of these species in conventional and preventative medicine has a long history. Their inherent biological activities, attributed to the presence of unique secondary metabolites, have enabled their utilization across a wider spectrum of therapeutic applications (Tepe, Cakir & Sihoglu Tepe, 2016; Erenler et al., 2018).

The commercial value of Origanum species further enhances their significance. Origanum onites L., commonly known as “Turkish oregano”, “Izmir oregano”, or “bilyalı kekik”, stands out as the most common species among oregano with respect to both production and trade volumes. Origanum is used to produce essential oils, spices, herbal teas, traditional medicine, and various beverages (Başer, 2022). Oregano has historically been a significant culinary herb in the Mediterranean region. Its characteristic aroma and flavor make it a versatile ingredient in numerous dishes, such as baked goods, vegetable dishes, legumes, fish preparations, pizzas, and pasta sauces. Beyond its culinary applications, oregano has also found its way into aromatic teas, and its presence is a frequent feature within Turkish cuisine (Singletary, 2023).

Numerous in-depth research studies have been conducted on spices and herbs due to their strong antioxidant capacity and the positive impact of their nutritional properties on human well-being. The incorporation of spices and various herbs into daily dietary patterns represents an effective approach for increasing antioxidant consumption, while simultaneously enhancing the intake of essential micro- and macro-elements, which may consequently lead to improved health outcomes. Antioxidants, comprising a diverse category of bioactive compounds present in spices, encompass flavonoids, phenolic compounds, sulfur-containing substances, tannins, alkaloids, phenolic diterpenes, and vitamins (Olukemi et al., 2005; Ulewicz-Magulska & Wesolowski, 2023; Kamiloglu et al., 2014). The antioxidant properties of these substances vary. Flavonoids, for instance, can neutralize free radicals and form complexes with catalytic metal ions, effectively rendering them inactive. Research has demonstrated that herbs and spices such as oregano, sage, and rosemary are excellent sources of antioxidants due to their high concentrations of phenolic compounds (Yashin et al., 2017).

Spice plants’ bioactive chemical composition varies with harvesting time, processing method, and post-harvest application, all of which can significantly influence the plant’s medicinal qualities. There are specific requirements for every species in order to get the right active chemical composition (Olukemi et al., 2005; Hazrati, Mousavi & Nicola, 2024).

This study investigates the impact of harvest timing on the antioxidant activity, levels of micro- and macro-elements, and heavy metal content of two widely used oregano species, Origanum onites L. and Origanum vulgare, as well as the endemic species Origanum acutidens L. In addition to selecting the most widely produced species in Türkiye, Origanum onites and Origanum vulgare, the endemic species Origanum acutidens was also included. O. acutidens thrives on sloping, sun-exposed hillsides with calcareous and chalky rocks, requiring relatively little moisture during its growing season. Furthermore, the plant is known to have been historically consumed as a spice, tea, and flavoring agent (Cosge et al., 2009; Yildirim, 2013). The appealing aroma of O. acutidens makes it a suitable candidate for ornamental applications in urban landscaping and rock gardens. Furthermore, its fragrant and disease-protective characteristics support its use in the food industry as a natural flavoring and preservative (Çoban, Özer & Cakmakcı, 2024).

This research aims to provide valuable insights on the optimization of harvesting times for oregano species, ensuring their maximum bioactive compound content, enhanced antioxidant activity, and optimal levels of micro- and macro-elements. Additionally, it evaluates the accumulation of heavy metals to ensure the safety and quality of these herbs. By understanding the interplay between harvest timing and these factors, one can harness the full potential of these herbs for medicinal purposes. Utilizing a range of oregano species, including endemic varieties, this study seeks to determine which species, under identical environmental pressures, exhibit the greatest antioxidant activity, dry herbage production, and balanced elemental composition.

MATERİALS & METHODS

Plant material

Origanum onites L., Origanum vulgare L. ssp. hirtum, and Origanum acutidens (Hand.-Mazz.) Ietswaart, the latter being an endemic species in Türkiye, were propagated from seed in 2020. The healthy seedlings were then transplanted into the experimental area of Yozgat Bozok University (39°45′09″N, 34°48′10″E, 1.266 m above sea level). Figure 1 presents the long-term climatic data (1990–2022) for the experimental site, along with the specific meteorological conditions during the harvest period. The soil properties of the experimental area were characterized by a high saturation percentage, indicative of substantial water-holding capacity. The soil texture was classified as silty clay loam. Chemical analysis revealed a slightly alkaline pH, negligible salinity, moderate levels of calcium carbonate, a high organic matter content, elevated levels of potassium, and moderate levels of phosphorus. The experiment was established in a Randomized Complete Block Design (RCBD) with three replications. Each plot, measuring 5 m in width and 2.5 m in length, consisted of five planting rows. A 1-meter gap was maintained between adjacent blocks. A drip irrigation system was set up for the experiment, and irrigation was performed during drought seasons. At planting, fertilization was performed at a rate of 6 kg N (as ammonium nitrate, AN) and 6 kg P2O5 (as triple superphosphate) per decare (1,000 m2). Half of the nitrogen was applied at planting, and the remaining half was applied in the spring. This fertilization regime was implemented only in the first year; no subsequent fertilization was carried out in the following years. Hand-hoeing was employed for weed control as necessary to maintain experimental conditions. Plant aerial parts were harvested after 3 years of adaptation to the field. Aerial parts were sampled from five plants per plot for each of the three oregano species at three flowering phases: pre-flowering (PF), early flowering (EF), and full flowering (FF). The developmental stages of the plants were determined according to the BBCH scale, as defined by Davidenco et al. (2015). Accordingly, the pre-flowering (PF) stage corresponded to BBCH stage R4, the early flowering (EF) stage to R5, and the full flowering (FF) stage to R6. For each flowering stage, morphological observations were conducted on five plants in each of three replicates. Following harvest, plant material was dried in the shade under controlled room conditions until a constant weight was reached.

Figure 1 Long-term climatic data (1990–2022) and meteorological conditions during the harvest period at the experimental site.

DPPH radical scavenging activity assay

The ability of oregano extracts to neutralize free radicals was investigated using the DPPH (2,2-diphenyl-1-picrylhydrazyl) assay, a widely employed method for evaluating antioxidant potential. The DPPH analysis was carried out using the procedure of Esmaeili et al. (2015) with small adjustments. The DPPH solution was prepared in methanol and adjusted to a concentration of 0.1 µM for the analysis. An extract stock solution was made at 1 mg mL−1 and diluted to obtain 6 different concentrations. A total of 3 mL of each concentration (10, 20, 40, 80, 160, 320) were withdrawn and 1 mL of 0.1 µM DPPH was added. The prepared samples were incubated in the dark for 30 min and compared to the blank. BHT (butylated hydroxytoluene) and BHA (butylated hydroxyanisole) served as reference antioxidants. Each sample was run in four replicates. The subsequent equation was employed to determine the DPPH radical scavenging activity:

% Inhibition = [(S0 −S1)/S0] × 100

where:

- S0 = Absorbance of the control sample at 517 nm

- S1 = Absorbance of the sample containing the extract at 517 nm.

Total phenolic content measurement (folin-ciocalteu assay)

Total phenolic content (TPC) is a key indicator of the antioxidant capacity of plant extracts. TPC method, known for its simplicity, reproducibility, and sensitivity, was utilized to measure TPC with modifications based on the methodology of Singleton, Orthofer & Lamuela-Raventós (1999). In this procedure, 40 µL of oregano extract was added to test tubes containing 2.4 mL of distilled water. For the control groups, 40 µL of methanol replaced the oregano extract. Following this, 200 µL of Folin reagent and 600 µL of saturated Na2CO3 solution were added into each test tube. The mixture was vortexed to ensure homogeneity and incubated at room temperature for 2 h. Absorbance was measured at 765 nm using a PerkinElmer Lambda 25 UV/VIS spectrophotometer. Gallic acid was used as the reference standard for quantifying phenolic compounds, with a calibration curve prepared using gallic acid solutions at concentrations of 50, 100, 150, 250, 350, and 500 µg mL−1. All experiments were conducted in quadruplicate to ensure the reliability of the results.

Total flavonoid content (aluminum chloride (AlCl3) assay)

Total flavonoid content (TFC) measurement assay was performed utilizing the procedure of Biju et al. (2014) with slightly modifications. In simply, 50 µl of 1 mg mL−1 oregano extract was added to test tubes, followed by 950 µl of methanol. Distilled water (4 mL) was added to ensure complete dissolution. Afterwards, 0.3 mL of 5% sodium nitrite (NaNO2) solution was added, and the mixture was reacted for 5 min. Subsequently, 0.3 mL of a 10% (w v−1) aluminum chloride (AlCl3) solution was added to the mixture, followed by incubation for 6 min at room temperature. Two mL of a 1 M sodium hydroxide (NaOH) solution was added following the incubation period. Distilled water (2.4 mL) was added to bring the final volume to 10 mL. A spectrophotometer was used to measure the absorbance of the mixture at 510 nm after it had been incubated for 15 min. A quercetin standard curve was constructed using six different concentrations (50, 100, 150, 250, 350, and 500 µg mL−1) prepared from a 1 mg mL−1 stock solution. The TFC was expressed as mg quercetin equivalents (QE) g−1 extract. Each experiment was performed in quadruplicate to ensure data reliability.

Determination of ferric reducing antioxidant potential (FRAP method)

The ferric reducing ability of plant extracts was assessed using the ferric reducing antioxidant power (FRAP) assay, adapted from the methodology described by Benzie & Strain (1996). A newly prepared FRAP reagent solution, combining acetate buffer (pH 3.6), 2,4,6-tripyridyl-s-triazine (TPTZ), and ferric chloride solution in a 10:1:1 ratio (v:v:v), was used. 40 µl of diluted extracts were added to 1.2 mL of FRAP reagent, incubated at 37 °C for 2 min, and the absorbance at 593 nm was recorded after incubation. A standard curve prepared with known FeSO4⋅7H2O concentrations (100, 250, 500, 750 1,000 µl mL−1) was used for calibration. The ferric reducing antioxidant capacity (FRAP) of the plant methanolic extract was determined using a calibration curve and expressed as micromoles (µmol) of ferrous sulfate (FeSO4) equivalents per liter (L).

Trolox equivalent antioxidant capacity assay: ABTS+ absorbance

Detailed methodology regarding Trolox equivalent antioxidant capacity (TEAC) assay was previously described by Re et al. (1999). A slightly modified method was employed in this study. The 2,2’-azino-bis(3-ethylbenzothiazoline-6-sulfonic acid) (ABTS) assay involved preparing an acetate buffer (pH 4.5) and separate ABTS and potassium persulfate stock solutions. These stocks were mixed (1:1) and incubated for 12 h, followed by dilution. The working solution was then reacted with samples (2 mL ABTS + 100 µl plant extract) for 7 min at 734 nm to assess their antioxidant capacity (target absorbance for working solution: 0.740 ± 0.030). A 2 mM Trolox stock solution is prepared by dissolving 1.0012 mg powder in 2,000 µL. Working solutions (12.5–400 µM) are then obtained by serially diluting the stock solution to achieve the desired concentrations. This ensures a linear absorbance response at 734 nm within the 12.5–400 µM Trolox range. The antioxidant capacity, reflecting the ability to quench ABTS radicals, was determined using the calibration curve and expressed as mg Trolox equivalent (TE) per mg of dry weight (DW) of oregano extract.

Micro-macro nutrient and heavy metal content (ICP-MS)

The micro- and macro-nutrient, as well as heavy metal content, were quantified using inductively coupled plasma-mass spectrometry (Thermo Scientific ICAPQC). The operational parameters were configured as follows: radiofrequency power at 1,550 W, nebulizer gas flow rate of 0.96 L min−1, plasma gas flow rate of 0.88 L min−1, nebulizer pressure at 3.01 bar, dwell time of 0.01 ms, and spray chamber temperature set to 3.7 °C.

To prevent contamination, the sampler probe was thoroughly cleaned between injections by rinsing with ultrapure water for 30 s, followed by a wash with 2% HNO3 for 45 s, and a final rinse with ultrapure water for 45 s. To ensure measurement precision, each sample and standard was analyzed in triplicate. A 0.5 g sample was digested in Teflon vessels with eight mL of suprapure nitric acid (HNO3) in a microwave digestion system (Milestone D5). After cooling, the clear supernatant was transferred to polypropylene tubes and diluted to 15 mL with deionized water. The concentrations were then adjusted for dilution and expressed in mg kg−1 as parts per million (ppm) or parts per billion (ppb).

Statistical analysis

Statistical analyses were conducted using a randomized complete block design (RCBD), and the data were tested using ANOVA that included species and harvest period as the main factor and block as a random effect. Assumptions of ANOVA were verified by testing the normality of residuals with the Shapiro–Wilk test and the homogeneity of variances with Levene’s test. Where significant treatment effects were detected, pairwise comparisons among group means were performed using the least significant difference (LSD) test. Statistical significance was considered at α = 0.05, and results were presented as mean ± standard deviation (SD). Principal component analysis (PCA) was performed using the Jump software package. Correlation analysis was conducted using the open-source software R. The corrplot function in R is employed to visually represent the correlation matrix, creating a graphical depiction of the relationships between variables (Murdoch & Chow, 1996; Hahsler, Hornik & Buchta, 2008).

Results

The plant characteristics of the studied species are presented in Table 1. All three species exhibit an erect growth habit. Only O. acutidens exhibited a more decumbent growth habit in the later stages of development. The fact that plant height in the species O. acutidens is higher during the early developmental stages and gradually decreases in the subsequent stages is also due to this reason. Consequently, the shortest plant height was observed in this species. Among the studied oregano species, the tallest plants were determined in the O. vulgare during the FF stage.

The studied species, the largest canopy diameter was observed in the O. vulgare during the EF and FF stages. The canopy diameter of O. acutidens was greater than O. onites, except during the PF stage (after the PF stage, the plant growth habit changes).

An evaluation of fresh herbage yield revealed that a significant difference among developmental stages was observed only in O. onites, whereas the differences among developmental stages in the other species were found to be statistically insignificant. In O. onites, the highest fresh herbage yield was recorded at the FF stage. Among the oregano species, the highest fresh herbage yield was observed in O. vulgare during the FF stage, followed by O. acutidens (FF) and O. vulgare (EF);however, the differences in fresh herbage yield among these species and stages were not statistically significant. The increase in fresh herbage yield was accompanied by an increase in dry herbage yield in most cases. However, the pattern differed between species. The highest dry herbage yield was determined solely for O. vulgare during the FF period, which statistically differed from other species and harvest periods.

The antioxidant properties of Origanum species, as determined by DPPH, ABTS, and FRAP analyses, are presented in Table 2, while the TPC and TFC values are presented in Table 3. The antioxidant activities of Origanum species are observed to be statistically significant both within species at different harvest periods and among different species. Among the extracts, O. acutidens exhibited the highest DPPH free radical scavenging activity during the EF developmental stage. O. onites exhibited the lowest antioxidant activity during the EF developmental stage. O. acutidens exhibited low antioxidant activity during the PF stage. Based on the FRAP assay results, O. vulgare and O. onites exhibited the highest antioxidant activity during the PF developmental stage. ABTS results also support that O. vulgare exhibits high antioxidant activity during the PF development stage. According to ABTS and FRAP assays, antioxidant activity in O. onites decreases from the PF to FF period. This finding suggests that the PF stage is a critical period for maximizing the antioxidant potential of O. onites. Therefore, harvesting O. onites during the PF stage may be particularly advantageous for obtaining extracts with high antioxidant activity.

Table 1 Evaluation of oregano plant yield and some agronomic properties across development stages.

Plant extract	Plant height (cm)	Canopy diameter (cm)	Fresh herbage yield (g)	Dry herbage yield (g)	
O. accuditens (PF)	40.11 ± 2.01a D1	43.11 ± 2.22b E	186.67 ± 38.39ns CD	38.77 ± 8.23b F	
O. accuditens (EF)	33.78 ± 1.17b E	59.44 ± 1.64a B	226.38 ± 32.11ns BC	76.72 ± 9.74a D	
O. accuditens (FF)	25.89 ± 1.02c F	60.67 ± 1.53a B	286.71 ± 26.76ns AB	95.57 ± 8.92a C	
LSD1(0.05)	3.70	4.36	–	23.53	
O. vulgare L. (PF)	48.22 ± 0.51c C	48.11 ± 1.26b CD	199.11 ± 11.0ns CD	62.55 ± 4.47c E	
O. vulgare L. (EF)	65.11 ± 4.86b B	75.33 ± 5.04a A	281.86 ± 9.82ns AB	116.94 ± 2.72b B	
O. vulgare L. (FF)	79.56 ± 0.84a A	75.00 ± 2.33a A	292.25 ± 83.31ns A	153.34 ± 2.71a A	
LSD1(0.05)	6.25	8.79	–	8.77	
O. onites L. (PF)	46.33 ± 1.67b C	50.00 ± 2.08ns C	142.89 ± 6.74b DE	55.48 ± 1.68b E	
O. onites L. (EF)	44.56 ± 3.28b CD	41.39 ± 4.57ns E	115.70 ± 6.93c E	38.57 ± 2.31c F	
O. onites L. (FF)	60.00 ± 4.22a B	51.89 ± 5.74ns C	185.86 ± 2.97a CD	77.81 ± 0.76a D	
LSD1(0.05)	9.91	–	12.54	3.88	
LSD2(0.05)	5.53	5.95	61.83	10.05	
Notes.

1The statistical significance of the difference between means denoted by the same letter is negligible. The first acronym represents an intraspecies comparison of statistical groups. The second, all-caps acronym indicates an interspecies comparison of characteristics. LSD1(0.05): Comparison within each species (among growth stages PF, EF, FF). LSD2(0.05): Overall comparison among all species and growth stages.

Table 2 Evaluation of DPPH antioxidant activity of Oregano extracts at various flowering stages.

Plant extract	DPPH SC50value1(µg mL−1 )	ABTS (mg TE mg −1 DW)	FRAP (µmol FeSO4 L −1 )	
O. accuditens (PF)	149.81 ± 0.40a D2	0.615 ± 0.023b G	700.93 ± 7.71c I	
O. accuditens (EF)	101.85 ± 1.65c H	0.670 ± 0.008a F	890.63 ± 8.59a F	
O. accuditens (FF)	115.81 ± 0.38b G	0.597 ± 0.003b G	827.40 ± 5.56b H	
LSD1(0.05)	2.00	0.028	14.78	
O. vulgare L. (PF)	142.72 ± 1.39b F	0.997 ± 0.001a A	1445.49 ± 21.87a B	
O. vulgare L. (EF)	152.95 ± 2.74a C	0.909 ± 0.009c B	1219.27 ± 15.19c D	
O. vulgare L. (FF)	145.88 ± 0.72b E	0.971 ± 0.013b B	1319.91 ± 16.69b C	
LSD1(0.05)	3.64	0.018	36.25	
O. onites L. (PF)	163.18 ± 0.87b B	0.964 ± 0.002a B	1497.15 ± 8.59a A	
O. onites L. (EF)	167.02 ± 0.63a A	0.849 ± 0.012b D	1053.62 ± 5.56b E	
O. onites L. (FF)	143.80 ± 0.43c EF	0.704 ± 0.018c E	870.15 ± 4.08c G	
LSD1(0.05)	1.34	0.025	12.71	
LSD2(0.05)	2.16	0.021	20.40	
Notes.

1The SC50 represents the concentration of the sample required to scavenge 50% of DPPH radicals; the data are expressed as mean ± standard deviation (SD), BHA SC50: 16.76 ± 0.47 μg mL−1, BHT SC50: 15.54 ± 1.75 μg mL−1

2The statistical significance of the difference between means denoted by the same letter is negligible The first acronym represents an intraspecies comparison of statistical groups. The second, all-caps acronym indicates an interspecies comparison. LSD1(0.05): Comparison within each species (among growth stages PF, EF, FF). LSD2(0.05): Overall comparison among all species and growth stages.

Table 3 Total phenolic content (TPC) of oregano extract.

Plant extract	TPC (mg g −1 GAE)	TFC (mg g −1 QE)	
O. accuditens (PF)	119.56 ± 11.23b C1	383.17 ± 20.89b C	
O. accuditens (EF)	158.63 ± 16.19a AB	640.39 ± 49.95a A	
O. accuditens (FF)	146.10 ± 18.40a BC	643.72 ± 49.49a A	
LSD1(0.05)	21.54	67.75	
O. vulgare L. (PF)	158.63 ± 16.19ns AB	442.33 ± 37.89ns B	
O. vulgare L. (EF)	191.63 ± 53.37ns A	397.33 ± 40.9ns BC	
O. vulgare L. (FF)	175.83 ± 21.9ns AB	444.28 ± 17.25ns B	
LSD1(0.05)	–	–	
O. onites L. (PF)	166.11 ± 10.74ns AB	373.17 ± 30.03ns C	
O. onites L. (EF)	168.42 ± 18.65ns AB	372.58 ± 14.20ns C	
O. onites L. (FF)	159.94 ± 13.28ns AB	377.61 ± 15.43ns C	
LSD1(0.05)	–	–	
LSD2(0.05)	33.4	48.85	
Notes.

1The first acronym represents an intraspecies comparison of statistical groups. The second, all-caps acronym indicates an interspecies comparison, TPC, Total phenolic content; TFC, Total flavanol content; Average of four analyses ± SD. Levels not connected by same letter are significantly different. LSD1(0.05): Comparison within each species (among growth stages PF, EF, FF). LSD2(0.05): Overall comparison among all species and growth stages.

The TPC and TFC content of oregano extracts are presented in Table 3. Based on Table 3, the highest TPC content was observed in O. vulgare during the EF period, while the lowest TPC content was found in O. acutidens during the PF period. Statistical analysis showed that only O. acutidens at both the PF and FF stages had significantly lower TPC contents compared to the other species and harvest periods. In the intraspecific comparison of TPC and TFC contents, statistically significant differences across developmental stages were observed only in O. acutidens. In O. onites and O. vulgare, changes in TPC and TFC contents according to developmental stage were found to be statistically insignificant.

The highest TFC content was observed in O. acutidens during the EF and FF stages. The TFC content in O. onites remained relatively stable throughout all developmental stages and was consistently lower than in the other species.

Agglomerative hierarchical cluster analysis and a correlation matrix were used to investigate antioxidant activity, agronomic characteristics, and mineral matter content in oregano species (Fig. 2). Figure 2 revealed the formation of three distinct groups. Group 1 consisted of potassium (K), phosphorus (P), TFC, fresh herbage yield (FHY), canopy diameter (CD), and dry herbage yield (DHY). Group 2 included plant height (PH), magnesium (Mg), zinc (Zn), TPC, boron (B), cadmium (Cd), calcium (Ca), ABTS, and FRAP. The third group comprised DPPH analysis and the elements copper (Cu), sodium (Na), manganese (Mn), nickel (Ni), arsenic (As), lead (Pb), Chromium (Cr), cobalt (Co), iron (Fe), aluminum (Al), and molybdenum (Mo). The attributes within each group demonstrated strong correlations and followed similar patterns. The correlation was strongly negative between TPC and K content. As K content increased, TPC decreased.

Figure 2 Correlation matrix and hierarchical cluster of antioxidant activity, agronomic characteristic and mineral matter content of oregano species (statistically insignificant correlations (p < 0.05) are represented by blank squares.

A strong positive correlation was found between FRAP and ABTS, while no significant correlation was observed between DPPH and these two methods.

A strong negative correlation was found between DPPH and TFC (Fig. 2). This means that as DPPH activity decreased (antioxidant properties increased), total flavonoid content also increased. A similar positive correlation was also observed between ABTS and FRAP. An increase in CD in plants leads to an increase in FHY, followed by an increase in DHY. This indicates that the increase in canopy diameter directly brings about an increase in yield and yield components.

According to Fig. 3, The biplot incorporates two principal components (PC), PC1 and PC2, which capture the most significant variation within the examined data set. In this case, PC1 accounts for 45.3% of the variation, while PC2 explains 37.7%. These two components together explain 83% of the total variation in plant characteristics and antioxidant activity observed across the oregano species and harvest times.

Figure 3 The biplot displays information about plant characteristics and the antioxidant properties of three oregano species harvested at three different stages (pre-flowering (PF), early flowering (EF), and full flowering (FF).

PC1 is most strongly influenced by PH, TPC, ABTS, FRAP, and DHY, all of which have high positive loadings. PC2, on the other hand, is primarily defined by high positive loadings for FHY, TFC and CD, and a strong negative loading for DPPH.

According to the PCA biplot, the most distinguishing features between the species are total flavonoid content (TFC) and DPPH radical scavenging activity. O. acutidens samples (1, 2, 3), especially at the full flowering stage (FF), are positioned in the upper left region, closer to the TFC vector, indicating higher total flavonoid content. O. onites samples (7, 8, 9) are located in the lower right region, particularly close to the DPPH vector, reflecting higher DPPH activity. In contrast, O. vulgare samples (4, 5, 6) are positioned in the middle-lower and lower-right regions of the biplot, not particularly close to any variable vector and relatively distant from both TFC and DPPH vectors. This suggests that O. vulgare is characterized by generally lower antioxidant and phenolic content compared to the other species. Thus, O. acutidens, O. onites, and O. vulgare are primarily separated in the PCA space by their relative positions to TFC and DPPH, as well as by their overall antioxidant and phenolic profiles.

The phenological stages within each species are also clearly differentiated in the PCA biplot. In O. acutidens, FF stage samples (3) are positioned closest to the TFC vector, while PF (1) and EF (2) stages are located more in the lower left region, further from TFC, indicating a progression in flavonoid content as flowering advances. For O. onites, PF (7) and EF (8) samples are located very close to the DPPH vector, showing high DPPH activity, whereas FF stage samples (9) are positioned higher and further from DPPH, suggesting a decrease in this activity at full flowering. O. vulgare samples at all phenological stages (4, 5, 6) are found relatively close to each other in the lower part of the biplot, distant from both TFC and DPPH vectors, with less pronounced differences between stages compared to the other species. This pattern demonstrates that phenological stages within all three species differ in terms of antioxidant and phenolic content, with O. acutidens showing the most significant changes in TFC, O. onites in DPPH activity, and O. vulgare exhibiting more subtle variations across flowering stages.

This study investigated the accumulation of macroelements (Ca, K, P, Mg), microelements (Fe, Zn, B, Mn, Cu, Na), and heavy metals (Al, Co, Cd, Cr, Ni, mercury (Hg), Pb, Mo, and As) in different oregano species at various harvest times. Statistical differences between species and among different harvest times within species, as well as the overall results of the study, are presented in Table 4 for macroelements, Table 5 for microelements, and Table 6 for heavy metals.

Across all three species, the most abundant macronutrients, based on their average values at different harvest times, were Ca, K, Mg, and P, respectively. The investigated macroelements were observed to accumulate in statistically different amounts both among species and within species at different harvest times, with the exception of P content within O. vulgare. This highlights the importance of both species selection and harvest time with regard to macroelement content. The highest values for Ca were determined in O. vulgare (PF and FF), for K and P in O. acutidens (PF), and for Mg in O. vulgare (FF).

Table 4 Macro-element content of Origanum species.

Macrominerals	
Species	Ca (ppm)	K (ppm)	P (ppm)	Mg (ppm)	
O. accuditens (PF)	10,084.66 ± 209.27c E1	12,092.85 ± 190.05a A	934.83 ± 1.56a A	3,140.75 ± 26.23c E	
O. accuditens (EF)	13,596.58 ± 283.39a C	9,059.09 ± 218.46b C	718.43 ± 11.17c C	3,515.18 ± 74.59a C	
O. accuditens (FF)	12,751.62 ± 71.87b D	8,724.43 ± 133,66b D	794.99 ± 9.58b B	3,318.77 ± 38.94b D	
LSD1(0.05)	414.73	367.87	17.07	101.66	
O. vulgare L.(PF)	16,260.27 ± 359.95a A	11,334.86 ± 131.98a B	768.88 ± 13.73ns B	3,598.42 ± 28.01b BC	
O. vulgare L. (EF)	14,207.63 ± 301.91b B	7,372.23 ± 269.30b F	793.52 ± 27.93ns B	3,560.11 ± 91.52b BC	
O. vulgare L. (FF)	15,818.10 ± 535.76a A	7,805.11 ± 275.81b D	787.41 ± 33.41ns B	4,359.86 ± 145.45a A	
LSD1(0.05)	821.93	469.99	–	195.95	
O. onites L. (PF)	13,837.77 ± 328.39a BC	8,721.72 ± 185,03a D	701.07 ± 17.41a CD	3,470.67 ± 88.12b C	
O. onites L. (EF)	12,769.92 ± 310.06b D	6,397.23 ± 156.86c G	608.74 ± 27.18b E	3,093.34 ± 69.47c E	
O. onites L. (FF)	12,563.24 ± 163.73b D	7,096.11 ± 99.72b F	670.88 ± 10.65a D	3,690.46 ± 84.35a B	
LSD1(0.05)	554.13	302.53	39.21	161.93	
LSD2(0.05)	532.66	331.70	33.63	137.46	
Notes.

1 The statistical significance of the difference between means denoted by the same letter is negligible. The first acronym represents an intraspecies comparison of statistical groups. The second, all-caps acronym indicates an interspecies comparison, ns: not significant (p < 0.05). LSD1(0.05): Comparison within each species (among growth stages PF, EF, FF). LSD2(0.05): Overall comparison among all species and growth stages.

Table 5 Micro-element content of Origanum species.

Micro-minerals	
Species	Fe (ppm)	Zn (ppm)	B (ppm)	Mn (ppm)	Cu (ppb)	Na (ppm)	
O. accuditens (PF)	144.49 ± 4.79a E1	13.49 ± 0.13c F	12.42 ± 0.86b G	39.66 ± 1.03a D	9,494.42 ± 83.08a B	35.62 ± 0.16b D	
O. accuditens (EF)	71.57 ± 1.51b H	13.99 ± 0.20b E	10.42 ± 0.87c H	33.58 ± 1.03b E	8,061.63 ± 154.62b E	9.77 ± 0.37c I	
O. accuditens (FF)	43.21 ± 0.88c I	15.11 ± 0.35a D	17.23 ± 0.82a F	25.45 ± 0.69c F	9,318.42 ± 177.15a B	39.01 ± 0.76a C	
LSD1(0.05)	5.88	4.90	1.70	1.86	287.66	0.99	
O. vulgare L.(PF)	229.12 ± 5.31a D	17.10 ± 0.29c C	40.30 ± 1.23a A	47.82 ± 1.22a C	8,552.77 ± 212.32b D	18.70 ± 0.31c H	
O. vulgare L. (EF)	84.85 ± 1.14c G	18.56 ± 0.17b B	30.70 ± 1.22c D	32.16 ± 0.46b E	7,620.46 ± 135.82c F	30.93 ± 0.70a E	
O. vulgare L.(FF)	125.78 ± 3.43b F	21.54 ± 0.47a A	35.24 ± 1.40b B	32.54 ± 0.98b E	9,381.19 ± 212.19a B	24.29 ± 0.66b G	
LSD1(0.05)	7.41	0.67	2.57	1.88	380.04	1.17	
O. onites L. (PF)	287.10 ± 5.81c C	12.88 ± 0.08c G	33.26 ± 0.41a C	47.48 ± 0.38c C	8,233.22 ± 14.58c E	27.78 ± 0.73c F	
O. onites L. (EF)	502.46 ± 8.77b B	13.47 ± 0.29b F	27.15 ± 2.08c E	52.96 ± 0.89b B	9,032.62 ± 148.06b C	100.18 ± 2.67a A	
O. onites L. (FF)	1105.86 ± 2.63a A	14.77 ± 0.24a D	30.62 ± 0.59b D	85.67 ± 0.57a A	1,0771.35 ± 106.33a A	94.98 ± 2.13b B	
LSD1(0.05)	12.51	0.44	2.54	1.30	210.94	4.03	
LSD2(0.05)	7.77	0.47	1.98	1.46	258.38	2.14	
Notes.

1 The statistical significance of the difference between means denoted by the same letter is negligible. The first acronym represents an intraspecies comparison of statistical groups. The second, all-caps acronym indicates an interspecies comparison, ns: not significant (p < 0.05). LSD1(0.05): Comparison within each species (among growth stages PF, EF, FF). LSD2(0.05): Overall comparison among all species and growth stages.

Table 6 Heavy metals content of Origanum species and WHO limits for edible plants.

Heavy metals	
Species	Al (ppm)	Co (ppb)	Cd (ppb)	Cr (ppb)	Ni (ppb)	Hg (ppb)	Pb (ppb)	Mo (ppb)	As (ppb)	
O. accuditens (PF)	400.25 ± 1.54a E1	179.44 ± 7.59a E	nd	1,189.45 ± 23.87a E	1,526.36 ± 80.53a FG	nd	181.87 ± 12.81F	339.01 ± 23.83b F	127.35 ± 6.83a DE	
O. accuditens (EF)	170.14 ± 5.50b G	95.90 ± 2.04b G	nd	579.97 ± 30.45b F	1.100.41 ± 42.85b H	nd	nd	1,199.05 ± 52.46a A	84.32 ± 19.03b FG	
O. accuditens (FF)	75.60 ± 1.03c H	73.47 ± 9.38c H	nd	415.30 ± 9.56c G	1,618.45 ± 16.24a EF	nd	nd	314,24 ± 8.70b F	57.50 ± 5.96c G	
LSD1(0.05)	6.70	14.12	-	45.97	106.88	-	-	67.22	24.31	
O. vulgare L.(PF)	639.62 ± 6.63a D	279.66 ± 10.98a D	12.85 ± 0.45ns	2,234.56 ± 70.28a D	1,722.97 ± 59.92b E	nd	366.36 ± 5.87a D	944.99 ± 10.18a C	161.77 ± 11.64a C	
O. vulgare L. (EF)	189.12 ± 5.76c G	123.89 ± 3.77c F	14.65 ± 4.65ns	696.73 ± 10.66c F	1,406.25 ± 34.61c G	nd	120.90 ± 4.43c G	635,00 ± 28.40c E	102.39 ± 7.45b EF	
O. vulgare L. (FF)	314.59 ± 9.51b F	168.52 ± 7.96b E	8.22 ± 1.60ns	1,296.43 ± 49.03b E	2,208.35 ± 43.80a D	nd	254.05 ± 12.75b E	698.47 ± 41.98b D	117.02 ± 18.07b E	
LSD1(0.05)	14.93	16.24	-	99.61	94.46	-	16.98	59.58	26.24	
O. onites L. (PF)	826.11 ± 17.04c C	371.25 ± 2.20c C	11.05 ± 0.00	3,050.08 ± 35.73c C	2,493.49 ± 32.82c C	nd	698.44 ± 3.26c C	942.07 ± 16.88c C	155.74 ± 16.68c CD	
O. onites L. (EF)	1,546.02 ± 62.33b B	668.34 ± 10.47b B	nd	5,967.01 ± 178.45b B	4,295.34 ± 173.04b B	nd	1,550.43 ± 23.85b B	1,236.00 ± 65.32a A	306.84 ± 16.68b B	
O. onites L. (FF)	3,541.48 ± 81.01a A	1,428.38 ± 19.81a A	12.59 ± 5.40	12,369.92 ± 144.65a A	7,325.52 ± 88.46a A	nd	3,676.45 ± 21.40a A	1,077.25 ± 30.69b B	490.49 ± 42.99a A	
LSD1(0.05)	119.54	25.97	-	268.16	227.34	-	37.16	85.50	53.53	
LSD2(0.05)	59.81	16.72	-	143.62	133.04	-	21.53	61.48	31.92	
WHO Limits	-	480∗	300∗	2,000∗	10,000∗	50∗∗	10,000∗	–	2,000∗∗	
Notes.

1 The differences between groups denoted by the same letter are not statistically significant. The first acronym represents an intraspecies comparison of statistical groups. The second, all-caps acronym indicates an interspecies comparison, WHO limits; ∗ Ssempijja et al. (2020), ∗∗Hu et al. (2023), nd: not detected, LoD (limit of detection): Cd: 0.50 ppb, Hg: 0.17 ppb, Pb: 1.12 ppb. LSD1(0.05): Comparison within each species (among growth stages PF, EF, FF). LSD2(0.05): Overall comparison among all species and growth stages.

The microelement content of the species also appears to be influenced by both harvest time and species. Regarding microelement content, O. onites (FF) exhibited the highest levels of Fe, Mn, and Cu; O. vulgare (FF) showed the highest level of Zn; O. vulgare (PF) had the highest level of B; and O. onites (EF) presented the highest level of Na. While O. onites did not exhibit prominence in terms of macroelement content, the highest accumulation levels of certain microelements were found in this species.

The analysis of heavy metal content revealed that the highest levels of Al, Co, Cr, Ni, Pb, and As were detected in O. onites (FF), whereas the highest level of Mo was found in O. onites (EF). These results suggest that, in plants grown in similar environments, O. onites tends to accumulate higher levels of many of the examined microelements and heavy metals, in contrast to macroelements.

Discussion

The results obtained in this study indicate that the height of oregano plants is generally consistent with similar studies in the literature conducted in different countries and conditions. Although different geographical regions and growing conditions can affect plant development, the findings align with the general trends in literature. O. onites are consistent with those previously reported by Kizil et al. (2008) and Kizil et al. (2009), while the results for O. vulgare align with the findings of De Mastro, Ruta & Marzi (2004) and Torres et al. (2010), and those for O. acutidens correspond with the observations of Karagoz et al. (2020).

Canopy diameter is an important parameter for oregano. A wider canopy diameter covers a larger area and provides shade while also aiding in water retention and ensuring that the plant benefits more from rainfall, particularly during drought periods. The greatest canopy diameter was observed in O. vulgare (EF and FF) and O. acutidens (FF). The lower DPPH activity, one of the total antioxidant measurements, observed in O. acutidens at the FF stage compared to the EF stage, as well as the higher antioxidant activity in O. vulgare at the PF stage compared to the EF and FF stages, may be associated with reduced water stress experienced by the plants, which is related to canopy diameter.

The fresh herbage yield of O. onites observed in this study aligns with the findings reported by Kizil et al. (2008). In contrast to Shokrgoo & Madandoust (2018) who found no effect of growth stage on plant height, fresh herbage yield, and dry herbage yield in O. vulgare, this study observed that different growth stages impact plant height and dry herbage yield but agree with their findings regarding fresh herbage yield. Significant differences were observed both among species and across developmental stages. Based on the comprehensive analysis of agronomic traits, O. vulgare demonstrated superior performance characteristics, suggesting its potential advantages for commercial cultivation. Dry matter accumulation is an important criterion, and it was determined that O. vulgare at the FF stage had nearly twice as much dry matter as O. onites at the FF stage.

The antioxidant properties of spices underscore their significance in both culinary applications and potential health benefits. The standard BHT and DPPH values of O. acutidens show parallels with the findings of Goze et al. (2010). The DPPH activity of O. onites is similar to that reported by Fidan et al. (2020a). According to the ABTS assay results, the greatest antioxidant activity was detected in the PF developmental stage of O. vulgare. The TPC and TFC content of O. acutidens is consistent with the findings of study Fidan et al. (2020b). The present study found higher total phenolic, DPPH, and FRAP values compared to those reported by Çıkla Yılmaz et al. (2019) using the O. onites samples collected from the natural environment. Although higher TFC and antioxidant values are generally expected in wild-collected samples due to increased environmental stress, this study found higher values in cultivated O. onites. This discrepancy may be attributed to differences in cultivation practices, soil and climatic conditions, harvest timing, or genetic variation. For O. vulgare, the TPC levels in this study were comparable to those reported by Karaboduk & Karabacak (2014), whereas the flavonoid content was significantly higher in this research. The variation in antioxidant activity based on both species and harvest time highlights the importance of optimizing harvest time for specific species and determining appropriate consumption periods for different species.

While Gaaliche et al. (2019) and Ibrahim et al. (2012) reported a positive effect of K fertilization on TPC in plants, Sahu et al. (2023) indicated that excessive K amounts negatively affected TPC. The strong negative correlation between TPC and K content may be related to the elevated K levels detected in the experimental soil, which could have affected phenolic compound accumulation in the plants. Some studies have shown a potential relationship between total phenolic content (TPC) and minerals such as P and N. However, it has also been noted that this relationship can be influenced by many factors, including the plant species, growing season, growing conditions, and even altitude (Chrysargyris, Evangelides & Tzortzakis, 2021). This study showed positive correlations in oregano species between TPC and Ca, and ABTS/FRAP with B, Cd, and Ca. In some stages of the examined species, Cd was detected above the limit of detection (LoD). This may be related to the presence of O. vulgare in all developmental stages and the generally high antioxidant activity in this species. Increased levels of antioxidant enzymes are known to effectively scavenge reactive oxygen species (ROS), thereby protecting cells from oxidative damage. Previous studies (Xu, Li & Zhang, 2013; Wang et al., 2009; Issam et al., 2012) have also demonstrated that external Ca2+ can significantly enhance the activity of key antioxidant enzymes such as SOD, POD, and CAT. In line with these findings, the observed positive correlation between Ca content and antioxidant capacity (as measured by ABTS and FRAP assays) in the present study suggests that Ca may contribute to cellular antioxidant defense mechanisms, potentially by promoting the activity of antioxidant systems.

The varying results obtained from different antioxidant assays, such as DPPH, FRAP, and ABTS, can be attributed to the diverse chemical structures of antioxidants and the distinct molecular targets of each assay. This finding is consistent with numerous studies in the literature, which have reported strong correlations between FRAP and ABTS but weak or insignificant correlations between DPPH and these methods (Rumpf, Burger & Schulze, 2023; Ramírez-García et al., 2022; Wootton-Beard, Moran & Ryan, 2011). These findings highlight the importance of considering multiple antioxidant assays when assessing the total antioxidant capacity of samples.

The uptake of various micro, macroelements, and heavy metas in plants can be interrelated. A negative correlation was observed between P and Mo. A similar negative correlation was also reported by Yamamoto et al. (2024). There is a statistically significant positive correlation among Cu, Na, Mn, Ni, As, Pb, Cr, Co, Fe, and Al. Minerals are vital for the human body’s healthy operation. Although they are vital for the upkeep and development of the body and are involved in a number of physiological processes, certain essential minerals, such as Zn, Cr, Hg, Pb, As, and Co,become toxic at high concentrations. Numerous facets of cell physiology and metabolism, such as stomatal opening, enzyme activation, cell proliferation, and turgor pressure maintenance, are influenced by K+. Plant cells keep an eye on the availability of K+ in roots to maintain a suitable K+ concentration (Shin, 2014). K acquisition exhibits variability across plant species. Nevertheless, plants typically absorb a predominant proportion of K during their early developmental phases compared to nitrogen and phosphorus (Prajapati, Arts & Collegekalol, 2012). Regardless of the species investigated, early harvest resulted in plants with the highest K levels. This is consistent with the general phenomenon of plants taking up the majority of their K during their early growth stages. Likewise, the fluctuations in P content over the development period and the lack of statistical significance for O. vulgare could be attributed to the same factors. Applications of Ca and Mg have a significant impact on plant height, dry herb yield, and fresh herb yield in O. vulgare (Dordas, 2009). In this study, the highest dry and fresh herb yield and plant height in O. vulgare were determined in the FF period, and the highest Ca (statistically the same group as PF) and Mg contents were also determined in this period. Among all three species, O. vulgare, which had the highest plant characteristics, also had the statistically highest Ca and Mg content. The findings of the present study regarding Ca content in O. vulgare are consistent with those of Dordas (2009), whereas the Mg content reported in that study is lower than the results obtained here. The findings of Amer et al. (2024) indicate that Zn application can stimulate plant height, fresh herbage and dry herbage yield in O. vulgare. The results of the present study corroborate these findings, as the highest Zn content and plant height were observed in O. vulgare during the FF period. This trend was consistent across all three species. In addition, Zn and Mn, applied alone or in combination, significantly increased Zn content and stimulated the production of biochemical components, including total phenolics, total flavonoids, and antioxidants. O. vulgare exhibited significantly higher Zn concentrations compared to other species across all harvest periods. Although TPC and ABTS values varied depending on the harvest period, they were found to be at higher levels compared to other species. The suboptimal antioxidant activity (DPPH, FRAP, TFC) observed in O. vulgare may be associated with the lower Mn levels detected in this species. Zn and Cu levels found in García-Galdeano et al. (2020) (collected from local shops) were similar to those found in this research. However, there were differences in Fe content among the species. Fe levels in O. acuditens (EF, FF) and O. vulgare (EF) were similar to their findings, but O. onites had higher Fe levels in all harvest periods. Similar to the findings of Thalassinos et al. (2024), no significant Cd content was detected in the leaves of O. vulgare and other oregano species in this study. Ait Bouzid et al. (2023) found that Ca, K, and Mn contents were slightly higher in their study on O. vulgare, while Na content was significantly higher than the values found in this study. The Zn content detected in their study was nearly twice as high as that found in the present study. Fe content exhibited significant variations across different species and harvest times. The content of Fe measured by Ait Bouzid et al. (2023) was comparable to the values observed in O. onites (FF). Likewise, the levels of B, Cu, As, and Mg were consistent with the results of the findings of this study. In terms of heavy metal contents compared to Ait Bouzid et al. (2023) study, Cd content was similar, while Pb content was higher in all harvest periods of O. vulgare and O. acutidens but higher only in the PF period of O. onites and lower in the EF and PF periods of O. onites. While the Hg content of oregano species could not be detected in this study, it was also found to be very low in the Ait Bouzid et al. (2023) study (Hg < 0.015 ppm).

Conclusion

The study revealed that species and harvest time significantly influenced key biochemical and nutritional parameters, including total antioxidant activity (DPPH, ABTS, and FRAP), TFC, TPC, and macro and micro-element, and heavy metals composition. Statistical analyses confirmed notable differences across species and harvest periods, underscoring the critical role of genetic and temporal factors in determining the quality and nutritional profile of oregano. These findings highlight the potential to optimize harvest timing and species selection to enhance desired antioxidant and nutrient characteristics for agronomic and commercial applications.

The study contributes to existing knowledge by providing a comprehensive evaluation of how species and harvest periods affect the nutritional and biochemical properties of oregano. This information is particularly valuable for producers and the agricultural industry, offering practical insights to improve the quality and marketability of oregano. However, the study is limited by its focus on specific species and harvest periods, suggesting the need for further research across a wider range of genotypes, environmental conditions, and cultivation practices.

Future research should explore the interaction between agronomic practices, environmental factors, and harvest timing to further optimize the antioxidant and mineral content of oregano. Speculatively, integrating precise harvest schedules with species-specific cultivation strategies could enhance the production of high-quality oregano with superior nutritional and commercial value. These findings provide a foundation for developing targeted approaches to improve oregano’s agronomic and nutritional potential.

Supplemental Information

Supplemental Information 1 The organized data used in the statistical analysis

Supplemental Information 2 The raw data used in the research

Supplemental Information 3 Codebook

Additional Information and Declarations

Competing Interests

Author Contributions

Data Availability

The author declares there are no competing interests.

Tansu Uskutoğlu conceived and designed the experiments, performed the experiments, analyzed the data, prepared figures and/or tables, authored or reviewed drafts of the article, and approved the final draft.

The following information was supplied regarding data availability:

The raw data is available in the Supplemental File.

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
