# Peer review of "Effect of harvest on the agronomic, mineral and antioxidant profile of three oregano species (Origanum onites L., Origanum vulgare L. ssp. hirtum, and Origanum acutidens (Hand.-Mazz.) Ietswaart)"

_PeerJ, doi:10.7717/peerj.20223_

## Round 0.1 · original submission · Major Revisions

**Language Note:** The review process has identified that the English language must be improved. PeerJ can provide language editing services - please contact us at [email protected] for pricing (be sure to provide your manuscript number and title). Alternatively, you should make your own arrangements to improve the language quality and provide details in your response letter. – PeerJ Staff

Reviewer 1 ·

Basic reporting

The manuscript entitled “Effect of Harvest on the Agronomic, Mineral and Antioxidant Profile of Three Oregano Species (Origanum onites L., Origanum vulgare L. ssp. hirtum, and Origanum acutidens (Hand.-Mazz.) Ietswaart)” is a manuscript that presents the differences in basic plant features between 3 different species of the origanum genus.

I see no aim and scope of this study, as it appears not as a research study, but as just an evaluation and comparison of 3 different species of the same genus, growing for 3 years, under unknown and undefined conditions, with no cultivation practices described.

The comparison of the features of the 3 different species might make some sense in cases, but how to compare different growth stages? Do these plants flower at the same time?

Experimental design

There is no experimental design in this study. There are no details presented not even for the sampling method.

Validity of the findings

-

Additional comments

Some extra notices:

Where is the accumulation of Nitrogen?

L29: What does “back many years” mean?

Why Essential oils have not been analyzed, a basic feature of the Origanum species.

In general, the discussion is just a description of the results compared with some already published data from the literature.

There is no critical aspect at all.

Reviewer 2 ·

Basic reporting

The manuscript investigates the effect of three harvest times (pre-flowering, early flowering, full flowering) on agronomic traits, mineral content (macro, micro, heavy metals), and antioxidant properties (DPPH, FRAP, ABTS, TPC, TFC) of three oregano species (Origanum onites, O. vulgare ssp. hirtum, O. acutidens) grown under identical ecological conditions in Yozgat, Türkiye. The study finds significant variations based on species and harvest time, suggesting potential for optimization. While addressing a commercially and scientifically relevant topic, several aspects could be improved for greater clarity and impact.
o Ecological Conditions Description: The study states plants were grown under "identical ecological conditions" at a specific location. However, details about these conditions (soil type analysis beyond mineral content, specific climate data during the growth/harvest season, irrigation regime, fertilization practices) are lacking. This makes it difficult to fully assess the environmental context and replicability.
o Harvest Stage Definition: The harvest stages (PF, EF, FF) are defined phenologically but lack precise quantitative criteria (e.g., percentage of flowers open for EF and FF). This could introduce variability.
o Replication Clarification: The methods state "five plants (with three replicates) of each of the three oregano species were sampled". It's slightly ambiguous whether this means 5 plants per replicate plot (totaling 15 plants per species/stage) or 5 plants total, analyzed in triplicate technically. Clarifying the experimental unit and replication structure (e.g., number of plots per species/treatment) is needed, especially concerning the use of RCBD vs. CRD for statistics.
o Antioxidant Assay Details: While standard methods (DPPH, FRAP, ABTS, TPC, TFC) are cited, some details could be clearer. For DPPH, the results are presented only as SC50 values, making direct comparison with other assays (often reported as TEAC or % inhibition at a specific concentration) difficult. Units for ABTS results are given as 'mg mg⁻¹ TE', which is unusual; typically, it's mg TE per g of extract or sample (e.g., mg TE/g DW). FRAP is expressed as 'µmol L⁻¹', likely meaning µmol FeSO₄ equivalents per L of extract, but should be explicitly stated or converted to per gram of sample for easier comparison.

Experimental design

o Statistical Test Justification: The manuscript mentions using RCBD for plant characteristics and CRD for antioxidant properties. The rationale for using different designs for different variable types measured on the same plants harvested at the same time is unclear and requires justification. Why wasn't a single design (likely RCBD if plots were blocked) used for all analyses?
o Interpretation of Significance: Some interpretations of statistical significance are confusing or potentially inaccurate. For example, the text states "Only in O. acutidens did developmental stages significantly influence TPC and TFC, while TPC and TFC changes were insignificant in other species". However, Table 3 shows 'ns' (not significant) for O. vulgare but provides different letters ('a', 'b') for TPC in O. acutidens, suggesting significance within that species was tested, but it doesn't explicitly confirm lack of significance within O. onites (also marked 'ns' but lacking letters). The comparison between species (capital letters) seems inconsistent in its application across tables.
o Correlation Interpretation: The correlation matrix (Fig 1) interpretation states a "strong negative correlation was found between DPPH and TFC". However, Figure 1 shows a positive correlation coefficient (around +0.2, though likely non-significant as it's in a blank box). It also states "as DPPH activity decreased (antioxidant properties increased), total flavonoid content also increased", which describes a positive correlation between TFC and antioxidant capacity (inverse of DPPH SC50), not the direct negative correlation stated initially. The relationship between K and TPC is discussed as negative, possibly due to "excessive K", but without soil K data, this is speculative.
o PCA Interpretation: The PCA explains 83% of the variance, which is good. The interpretation focuses on general positions (e.g., O. vulgare FF in upper right) but could be more specific about which variables strongly load onto PC1 and PC2 and how this drives the separation of species/harvest times. For instance, which specific traits most distinguish O. acutidens FF from O. onites FF?

Validity of the findings

o Language and Phrasing: Some phrasing is awkward or unclear, e.g., "displayed the lowest antioxidant activities back many years" - likely intended to mean "compared to previous years" or simply "very low", but confusing as written. The abstract and conclusions reiterate findings somewhat repetitively.
o Table Formatting and Clarity: The tables contain a large amount of data. The use of two sets of significance letters (lowercase within species, uppercase between species) is a valid approach but needs consistent application and clear explanation in every table footnote. Table 6 has inconsistent formatting in the column headers (some superscripts, some not) and includes WHO limits derived from multiple different sources, denoted by asterisks referring to references [45-47], which could be presented more clearly. The use of 'nd' (not detected) should specify the detection limit.
o Figure Quality: Figure 1 (correlation matrix) uses blank boxes for insignificant correlations but also has crossed-out circles in the description, which is inconsistent. Figure 2 (PCA biplot) is informative but could be slightly larger or use clearer labels for better readability.
o Comparison with Literature: The discussion compares findings with several previous studies. However, it sometimes just states agreement or disagreement without deeply exploring potential reasons for discrepancies (e.g., different environments, genotypes, specific assay variations). The discussion on heavy metals focuses heavily on comparing levels to WHO limits but less on the biological implications or potential sources in the specific experimental location.
o Rationale for Species Selection: The introduction mentions the diversity of Origanum in Türkiye but could provide a stronger justification for selecting these specific three species, particularly the endemic O. acutidens, beyond just including it. What unique properties were hypothesized or are known?
o Linking Antioxidants and Minerals: The study measures both antioxidant profiles and mineral content but doesn't explore potential links between them in the discussion (e.g., do micronutrients like Mn, Zn, Cu, known cofactors for antioxidant enzymes, correlate with measured antioxidant activity?).

Additional comments

The manuscript provides valuable data on the influence of species and harvest time on oregano's agronomic, antioxidant, and mineral characteristics. The finding that O. vulgare generally showed higher yields and O. acutidens or O. vulgare often exhibited higher antioxidant markers at specific stages is useful. However, the study's rigor and clarity could be enhanced by providing more detailed methodological context (environmental conditions, replication), refining the statistical analysis presentation and interpretation (design justification, significance notation, correlation accuracy), improving language and figure/table clarity, and deepening the discussion regarding mechanistic links and reasons for discrepancies with literature. Addressing these weaknesses would significantly improve the manuscript's quality and impact.

Reviewer 3 ·

Basic reporting

Introduction Section
2nd paragraph
Some sentences contain repetition and indirect constructions; clearer and more concise expressions would be preferable. Phrases such as “finds applications in...” could be replaced with stronger and more precise verbs. Expressions like “even chilis” lean towards colloquial language — using more neutral or academically appropriate examples would improve the tone. I suggest that the authors consider revising the manuscript to improve the clarity and fluency of the English language. For example “ Türkiye has a very diverse…….”
3rd paragraph
The text is generally quite suitable for an academic paper. However, some minor adjustments would be more appropriate to enhance the use of a more academic and professional language. Below are some suggestions to improve the text. Flow of Language: The text is quite clear and structurally coherent, but certain sentences can be rearranged for a more academic tone. Verb Usage: Especially the phrase “have conducted” could be replaced with passive structures or more precise verbs. For example, “Numerous studies have conducted in-depth research” could be changed to “Numerous studies have been conducted on...” which is more common and academic. Avoiding Repetition: There are some repeated phrases, which can be simplified to avoid redundancy.
Paragraph 4
This text is generally suitable for academic paper language, but it can be made more academic and professional with the following minor adjustments. "Spice plant" vs. "spice plants": Using "spice plants" is more appropriate as it reflects the diversity among species. "May have a significant impact" vs. "can significantly influence": The latter is a stronger expression and sounds more academic. "Certain requirements" vs. "specific requirements": "Specific" is clearer and more precise in academic writing. Sentence structure: The revised version shortens and clarifies the sentence structure, making it more direct and readable.
Paragraph 5 and 6
It can be expressed more effectively with some minor word changes. For example: "Explores" vs. "investigates", "Impact" vs. "effects", "Further" vs. "Furthermore" and "Vary" vs. "fluctuate". "Provide insightful information" vs. "provide valuable insights", "Interplay of harvest timing" vs. "interplay between harvest timing and these factors", "Demonstrate" vs. "exhibit", Including endemic varieties" vs. "utilizing a range of oregano species, including endemic varieties".
Materials & Methods
I suggest that the authors consider revising the manuscript to improve the clarity and fluency of the English language. For example “..Ietswaart (endemic species in Türkiye) were propagated… ” in 2nd line.

Results and Discussion
1st paragraph
For improved clarity and readability, it may be helpful to ensure consistency in verb tense, refine word choice, and incorporate more precise scientific terminology where appropriate. And the plant height results have been compared with only a single species. However, it would strengthen the study if these findings were supported by more recent research investigating the agronomic characteristics of the species under study.
2nd paragraph
The substitution of certain words is recommended for better alignment with academic writing conventions, as outlined below. “more area vs. a larger area”, "especially during drought periods" vs. "particularly during drought periods" , "the highest canopy diameter was observed" vs "the largest canopy diameter was observed" and "was higher than" vs. "was greater than":

Experimental design

Upon reviewing the manuscript, it appears that the study is based on a two-factor experimental design—the first being different plant species, and the second being different flowering stages. However, this design is not mentioned in the Materials and Methods section, and the analysis appears to have been conducted as if it were a single-factor experiment. Despite this, the results are interpreted both across species and according to flowering stages. Such interpretation would only be valid if species and flowering periods were statistically evaluated together in a factorial design.

Since it is unclear whether there is a statistical interaction between plant species and flowering stages in terms of the traits examined, the lack of factorial analysis prevents accurate interpretation. The statistical analyses should be conducted using a two-factor design. If no interaction is found between the factors, grouping should be done separately for species and flowering stages. The current approach is incomplete, insufficient, and may lead to misinterpretation of the findings.

Validity of the findings

In addition, while individual plants were evaluated, there is no information provided on row spacing or plant spacing. Since plant growth can vary depending on planting density, more detailed information on cultivation practices is necessary. The plant has economic significance, and the primary harvested part is the leaf. However, the study mentions that the above-ground parts were used. When reporting dry herbage yield, there is no information on the drying temperature or method used. The study would have been more relevant had it focused on the commercially important leaf part.
Some suggestions have also been provided regarding language and structure.

The analyses conducted are valuable, but it is recommended that the statistical evaluations be revised according to the appropriate experimental design and interpreted using updated literature. In its current form, the manuscript is not suitable for publication.

Additional comments

here appears to be a statistical flaw in the formulation and evaluation of the research hypothesis. The study should either be re-evaluated as a two-factorial experiment or be limited to a single plant species to maintain experimental consistency. Additionally, the Materials and Methods section lacks sufficient detail and should be more comprehensively described.

The manuscript gives the impression that the study is divided into two disconnected parts, and it appears to omit key parameters such as essential oil content, essential oil yield, and dry leaf yield, all of which are critical in oregano research. While the topic is promising and the study has potential, it is currently technically insufficient in its present form.

Furthermore, the writing lacks clarity and strength in academic language. The discussion should be expanded and supported with more recent and relevant literature to enhance the scientific depth of the manuscript.

---

## Round 0.2 · Minor Revisions

Dear authors,
Thank you for resubmitting your manuscript for review. Two independent experts have evaluated your work. Based on their comments and a thorough assessment of the submitted article, we have asked for minor revisions before publication.

We strongly encourage the authors to once again revise and resubmit the manuscript after addressing the issues mentioned by Reviewer 3.

We are excited to receive the revised version of the manuscript, along with a detailed response to all reviewer and editorial comments.
With best regards,

Reviewer 2 ·

Basic reporting

The authors have implemented the necessary corrections and revisions.

Experimental design

-

Validity of the findings

-

Reviewer 3 ·

Basic reporting

The review has been completed, and the points that need to be addressed have been clearly indicated.

Experimental design

-

Validity of the findings

“ Among the oregano species, the highest fresh herbage yield was observed in O. vulgare during the FF stage, followed by O. acutidens (FF) and O. vulgare (EF).” It should be explicitly stated that all three specified plant groups fall within the same statistical group (A=AB), indicating no significant statistical difference among them.

“The increase in fresh herbage yield was accompanied by an increase in dry weight yield, and the highest dry herbage yields were also determined in the FF periods of the examined species. Similar to fresh herbage yield, the highest dry herbage yield was determined for O. vulgare during the FF period.” It appears there is no similarity between dry forage yield and green forage yield. Statistically, the highest dry forage yield was obtained solely from O. vulgare L. during the FF period.

“Based on the FRAP assay results, O. vulgare and O. onites exhibited the highest antioxidant activity during the BF developmental stage.” According to Table 2, the highest antioxidant activity was observed in O. onites during the BF stage. However, this significant finding has not been discussed in the manuscript. It would be beneficial to include this important result in the article.

The TPC and TFC content of oregano extracts are presented in Table 3. Based on Table 3, the highest TPC content was observed in O. vulgare during the EF period, while the lowest TPC content was found in O. acutidens during the BF period. According to Table 3, the highest TPC is not exclusive to O. vulgare. Statistically, it also applies to other groups classified as AB. It's important to clarify that while A=AB, A is not equal to B.

Additional comments

This is a valuable manuscript that would be suitable for publication once the necessary revisions have been completed.

---

## Round 0.3 · Minor Revisions

Dear Dr. Uskutoğlu,

Thank you very much for submitting the latest version of your article. After thoroughly reviewing the changes you made following the previous feedback, we would like to ask you to correct some shortcomings:

1) the type of statistical test used to compare means needs to be described in the footnote of tables 1-6 and/or in the Methods section. Presumably, ANOVA was conducted and post-hoc tests were then done to compare means, but there are many types of post-hoc tests and the specified test that was chosen should be used. If ANOVA or GLMs were used, these assume normally distributed residuals in order for inferences based on the post-hoc tests to be valid. However, you did not mention whether residuals were normally distributed or whether transformations were used.

2) regarding the Abstract, the acronyms DPPH, FRAP, and ABTS should be spelled out on first use. L 48 "In Turkish flora" -- please revise to "In the Turkish flora".

3) regarding Figure 3, Correlations between 8 minerals ranging between Al and Mn are all 1 (perfectly correlated). Please check this as, this seems unlikely.

4) it is not necessary to show both sides of a correlation matrix (above and below the diagonal) because each side shows identical information. But if you prefer to show values above and below the diagonal, then it is okay to keep it as is.

With best regards,

---

## Round 0.4 · Minor Revisions

Dear Author,

Please address these remaining items, noted by one of the Section editoris:

> The type of statistical test used to compare means needs to be described in the footnote of tables 1-6 and/or in the Methods section. Presumably, ANOVA was conducted and post-hoc tests were then done to compare means, but there are many types of post-hoc tests and the specified test that was chosen should be used. If ANOVA or GLMs were used, these assume normally distributed residuals in order for inferences based on the post-hoc tests to be valid. However, the authors did not mention whether residuals were normally distributed or whether transformations were used. 

> Regarding the Abstract, the acronyms DPPH, FRAP, and  ABTS should be spelled out on first use.

> L 48 "In Turkish flora" -- please revise to "In the Turkish flora".

> Regarding Figure 3, I noticed that correlations between 8 minerals ranging between Al and Mn are all 1 (perfectly correlated). Please check this, as this seems unlikely. Also, it is not necessary to show both sides of a correlation matrix (above and below the diagonal)  because each side shows identical information. But if the author prefers to show values above and below the diagonal then ok to keep as is.

---

## Round 0.5 · accepted · Accept

Dear Dr. Uskutoğlu,

I’m pleased to inform you that the revised manuscript “Effect of Harvest on the Agronomic, Mineral, and Antioxidant Profile of Three Oregano Species (Origanum onites L., Origanum vulgare L. ssp. hirtum, and Origanum acutidens (Hand.-Mazz.) Ietswaart)” (114913) has been reviewed. The author’s responses to the prior comments, provided in the rebuttal letter, have been approved. The manuscript has successfully addressed all the requested revisions and can be accepted for publication in PeerJ in its current form.